# Novel (*S*)-Selective Hydrolase from *Arthrobacter* sp. K5 for Kinetic Resolution of Cyclic Amines

**Yuta Fukawa [1], Yuta Mizuno [2], Keisuke Kawade [2], Koichi Mitsukura [2,*] and Toyokazu Yoshida [2]**

[1] Department of Engineering Science, Graduate School of Engineering, Gifu University, 1-1 Yanagido, Gifu 501-1193, Japan; y3921018@edu.gifu-u.ac.jp

[2] Department of Chemistry and Biomolecular Science, Faculty of Engineering, Gifu University, 1-1 Yanagido, Gifu 501-1193, Japan; aes128.111@docomo.ne.jp (Y.M.); hornet4243@yahoo.co.jp (K.K.); toyosida@gifu-u.ac.jp (T.Y.)

\* Correspondence: mitukura@gifu-u.ac.jp

**Abstract:** Chiral 2-methylpiperidine (2-MPI) is an important building block that has potential for applications in pharmaceuticals and pesticides. In this study, we observed that the hydrolase in *Arthrobacter* sp. K5 exhibits high (*S*)-selectivity toward *rac-N*-pivaloyl-2-MPI to yield (*S*)-2-MPI with 80.2% enantiomeric excess (*ee*) in a 38.2% conversion. The hydrolase, which was identified by analyses of partial amino acid sequences of the purified enzyme and genome sequence of *Arthrobacter* sp. K5, exhibited moderate homology with amidohydrolases up to 67% (molinate hydrolase from *Gulosibacter molinativorax*). The hydrolase gene was overexpressed in *Rhodococcus erythropolis*. The recombinant cells produced (*S*)-2-MPI with 83.5% *ee* in a 48.4% conversion (*E* = 26.3) from 100 mM *rac-N*-pivaloyl-2-MPI. These results suggest the possibility of an efficient preparation of chiral 2-MPI in kinetic resolution.

**Keywords:** hydrolase; *Arthrobacter* sp. K5; kinetic resolution; cyclic chiral amine

## 1. Introduction

Chiral cyclic amines, which comprise a piperidine, piperazine, or pyrrolidine skeleton, are widely used as building blocks of pharmaceuticals and pesticides [1–9]. Among them, 2-methylpiperidine (2-MPI) is used for the synthesis of piperocaine [6], piperalin [7], menabitan [8], SS220 [9] (Figure 1), and promising candidates for developing pharmaceuticals [3–5]. In the synthesis of SS220, (*S*)-2-MPI moiety makes it effective or more effective than the most widely used insect repellent, *N,N*-diethyl-*m*-toluamide (DEET) [10–12]. The enzymatic synthesis of optically active cyclic amines has been successfully done via asymmetric reduction using imine reductases [13–16]; however, the method requires the synthesis of cyclic imines as substrates. Kinetic resolution is another approach for preparing chiral compounds, in which the racemates are used as substrates. Since racemic cyclic amines can be commercially available as inexpensive raw materials, in this study, we focused on stereoselective hydrolases for 2-MPI production.

**Figure 1.** Biologically active compounds containing 2-MPI moiety.

Optically active amines can be prepared from racemic amine by diastereomeric salt formation using chiral carboxylic acids [17] or by stereoselective bioconversion of racemic *N*-acyl amine using enzymes [18–20]. Several studies on chiral cyclic amines' preparation via chemical or enzymatic resolution have been reported (Scheme 1) [21–33]. In the enzymatic method, racemic piperidines are stereoselectively acylated or hydrolyzed by a lipase or protease to obtain the desired enantiomer [21–24]. Conversely, the racemates are chemically resolved by chiral acylating reagents to provide enantioenriched amines [25–27]. Other chemical resolutions have been reported [28–30]; however, these chemical processes require chiral reagents for kinetic resolution. Considering the need of chiral compounds that are also synthesized from chiral precursors, the discovery of highly stereoselective hydrolase is desired for chiral cyclic amines' preparation. Until now, the existing hydrolases described above have been used to synthesize chiral secondary amines; however, their variation was limited. Other enzymatic processes need cyclic amines with functional groups, such as 2-hydroxymethylpiperidine, to yield products with low enantioselectivity in many cases due to the low chirality-recognition ability of the enzyme [31–33]. Interestingly, there is no approach to directly obtain chiral 2-substituted piperidines from racemic *N*-acyl derivatives by kinetic resolution using a hydrolase. Given these backgrounds, it is necessary to improve the enantioselectivity of existing hydrolases or find a novel hydrolase.

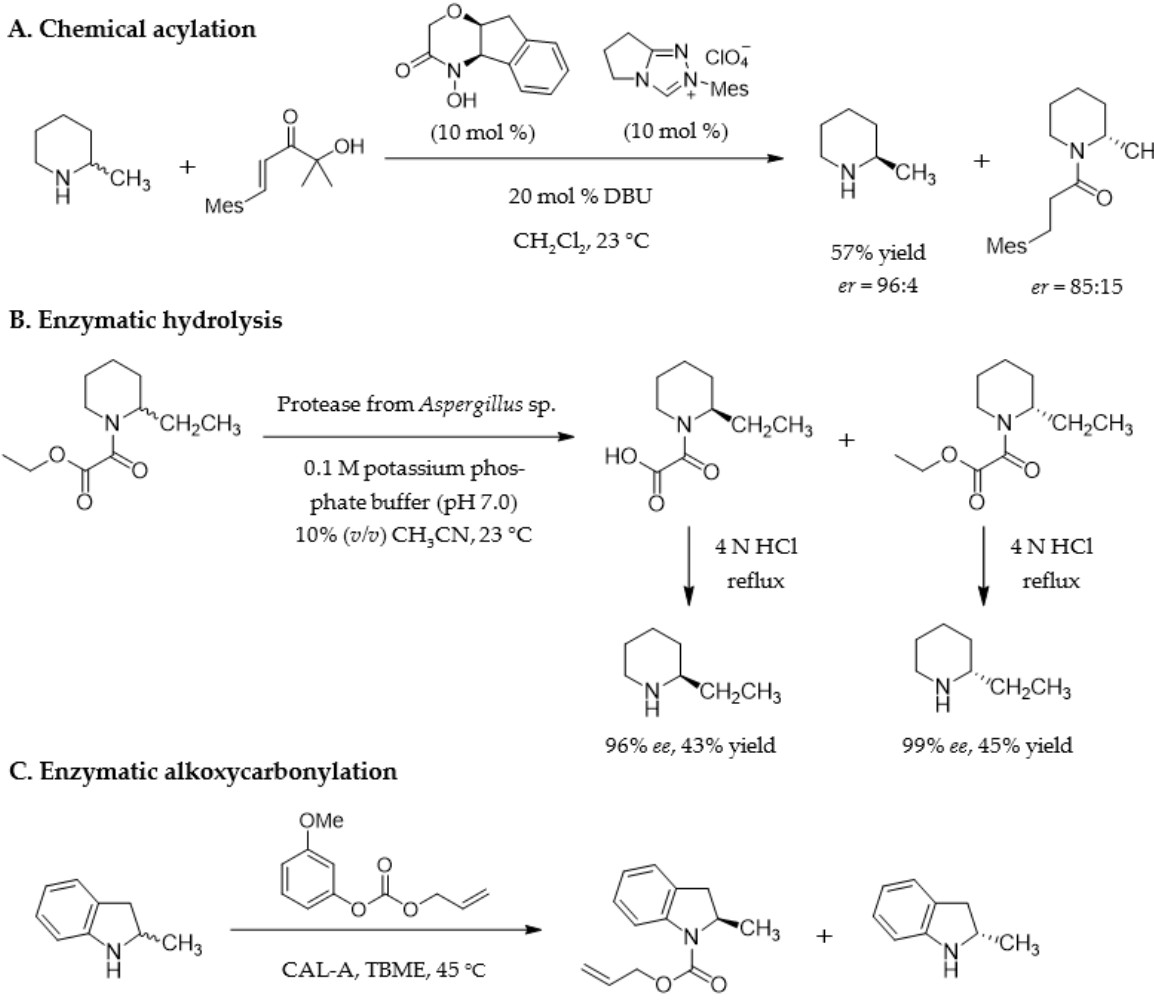

**Scheme 1.** Examples of reported processes for the preparation of chiral cyclic amine.

In this study, we screened the stereoselective hydrolases acting on *rac-N*-acyl-2-MPI from soil microorganisms for one-step preparation of chiral 2-MPI (Scheme 2). We then

purified and characterized the hydrolase of *Arthrobacter* sp. K5. To ensure efficient chiral piperidine preparation, we further constructed recombinant cells expressing the hydrolase gene and investigated the optically pure 2-MPI production via whole cells reaction.

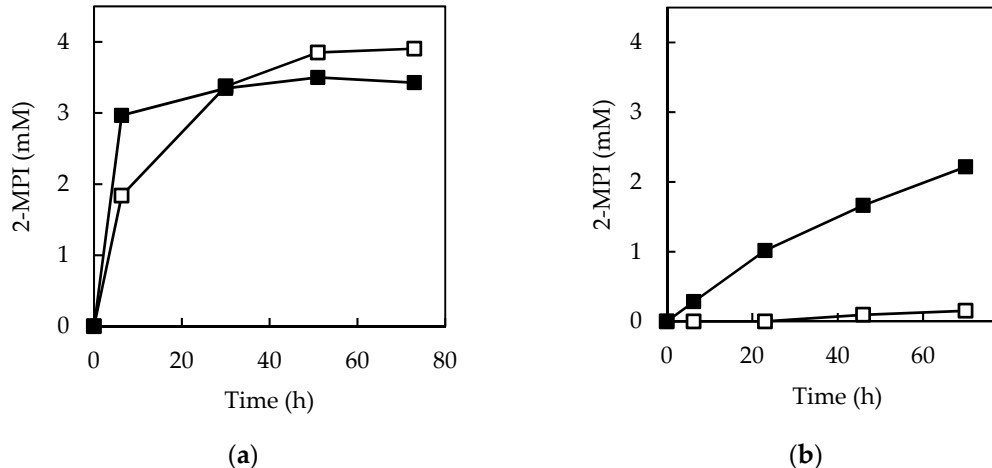

**Scheme 2.** Stereoselective hydrolysis of *N*-pivaloyl-2-MPI by *Arthrobacter* sp. K5 hydrolase.

## 2. Results and Discussion

### 2.1. Screening of Microorganisms

　　To obtain microorganisms exhibiting hydrolase activity toward *N*-acyl-2-MPI, we isolated 294 strains from soil samples using a medium containing *N*-acetyl-2-MPI as the sole carbon source. Among 60 bacteria that hydrolyzed *N*-acetyl-2-MPI, the majority of them did not display high stereoselectivity. Only the strain K5 revealed moderate (*S*)-selectivity kinetically, yielding racemic 2-MPI (Figure 2a). We predicted that the acyl group affected the enantiomeric recognition of the hydrolase in strain K5. To improve the enantioselectivity of the hydrolysis, we substituted the acetyl group with a bulky pivaloyl group for appropriate chiral recognition. The reaction using whole cells of strain K5 exhibited good (*S*)-stereoselectivity toward *N*-pivaloyl-2-MPI to yield (S)-2-MPI with 22% conversion and 88% *ee* in 72 h (Figure 2b). The strain K5 was identified as *Arthrobacter* sp. based on 16S rDNA sequence analysis.

**Figure 2.** Hydrolysis of *N*-acetyl-2-MPI (**a**) and *N*-pivaloyl-2-MPI (**b**) using *Arthrobacter* sp. K5 cells. Closed squares; (*S*)-2-MPI, open squares; (*R*)-2-MPI. The reactions were performed at 30 °C in 2 mL of 100 mM potassium phosphate buffer (pH 7.0) containing 10 mM *N*-acyl-2-MPI and whole cells derived from 4 mL culture broth.

### 2.2. Optimization of Culture Conditions and Kinetic Resolution Using Optimized Whole Cells

　　To enhance the hydrolase activity in *Arthrobacter* sp. K5 cells, cyclic amine derivatives were added to the culture medium, and the hydrolase activity was examined. The addition of *N*-acetyl-2-MPI, *N*-acetylpiperidine, or *N*-acetyl-2-methylpyrrolidine resulted in a high induction of the hydrolase activity, with the best compound being *N*-acetylpiperidine, which also enhanced *Arthrobacter* sp. K5 growth (Table S1). The highest activity was found in the cells cultivated for one day with 0.4% (*v/v*) *N*-acetylpiperidine (Tables S2 and S3). Prolonged cultivation (≥2 days) decreased enzyme activity. The culture conditions were

optimized as follows: time, 24 h and culture medium, 0.4% (*v/v*) *N*-acetylpiperidine, 5 g L$^{-1}$ of polypeptone, 5 g L$^{-1}$ of meat extract, 2 g L$^{-1}$ NaCl, and 0.5 g L$^{-1}$ yeast extract in tap water (pH = 7). Using the cells cultivated under optimal conditions, we performed kinetic resolution of *N*-pivaloyl-2-MPI. In the reaction with a 100 mM concentration of *N*-pivaloyl-2-MPI, *Arthrobacter* sp. K5 cells achieved 38.2% conversion to (*S*)-2-MPI at 80.2% *ee* (*E* = 14.8) after 115 h. However, the whole cells reaction stopped after 75 h, unable to achieve 50% conversion of the substrate. This result suggested the possibility of the enzyme inactivation caused by a long-time reaction or insufficient stability of the hydrolase.

## 2.3. Properties of the Hydrolase

To obtain a homogeneous enzyme, the hydrolase was purified from the cell-free extract of *Arthrobacter* sp. K5 through five steps, including ammonium sulfate fractionation, ion-exchange chromatography on DEAE-Sephacel, and hydrophobic interaction chromatography on phenyl-Sepharose and butyl-Toyopearl (Table S4). The purified enzyme was obtained with a specific activity of 35.5 μmol min$^{-1}$ mg$^{-1}$. The overall purification was 10.8-fold (yield = 22%). The molecular mass of the hydrolase was estimated to be 50 kDa by SDS-PAGE and 238 kDa by gel permeation high performance liquid chromatography (HPLC), which suggested it to be a homo-tetrameric hydrolase. The optimum reaction temperature and pH of the hydrolase were 45 °C and pH = 8.0 (Tris-HCl), respectively (Figures S2 and S3). The enzyme retained 75% of its maximum activity <40 °C; however, its activity decreased at a temperature >45 °C (Figure S4). Conversely, it retained 80% of its maximum activity at pH = 6.0–7.5 and more than 80% inhibition of the hydrolase activity was observed at other pH values (Figure S5). According to amino acid sequence analysis, the N-terminal and internal amino acid sequences of the hydrolase were obtained as ATQTVITNGTLIDGTGNQPQ and GGVTTVFDTWNA, respectively. Based on the amino acid sequence and genome DNA sequence of *Arthrobacter* sp. K5, we identified (*S*)-selective hydrolase (SHA). The enzyme gene was composed of 1384 bp, and coded for a protein of 481 amino acids with the molecular mass of 49,725 Da. This value is in agreement with molecular mass determined on SDS-PAGE. A BLAST search with full-length amino acid sequence of SHA revealed moderate sequence identity with the amidohydrolase protein family, including phenylurea hydrolases (<66%), and the highest sequence identity (67%) with the molinate hydrolase from *Gulosibacter molinativorax*. These results suggested SHA as a novel enzyme.

## 2.4. Substrate Specificity

We examined the substrate specificity of purified SHA using various *N*-acyl cyclic amines and comparing the activity toward them with the activity toward *N*-benzoyl-2-MPI (Table 1). The hydrolase exhibited almost the same activity toward *N*-benzoyl-2-MPI and *N*-pivaloyl-2-MPI; however, the latter was more hydrolyzed with higher (*S*)-selectivity. *N*-Acetyl-2-MPI was hydrolyzed with 24.8-fold higher activity than *N*-benzoyl-2-MPI. SHA displayed high activity on *N*-acyl 2-unsubstituted cyclic amines, such as *N*-benzoylpiperidine, *N*-benzoyl-3-MPI, *N*-pivaloyl-3-MPI, *N*-benzoyl-4-MPI, and *N*-benzoylpyrrolidine but showed no stereoselectivity toward *N*-acyl-3-MPI. *N*-benzoyl-2-methylpyrrolidine displayed four-fold higher reactivity than *N*-benzoyl-2-MPI, whereas enantioselectivity was low in slight favor of the (*R*)-enantiomer (11% *ee*). *N*-benzoyl-2-methylindoline was also a preferable substrate, reacting with medium enantioselectively (*S*- or *R*-enantiomers not determined); however, *N*-pivaloyl-2-methylindoline and *N*-benzoyl-1,2,3,4-tetrahydroquinaldine were not hydrolyzed. The reactivity and stereoselectivity of SHA for tested compounds depended on the acyl groups and the distance between the acyl group and chiral center. SHA exhibited no activity toward *N*-acetyl D- or L-amino acid (data not shown).

**Table 1.** Substrate specificity of SHA [1].

| Substrate | Relative Activity (%) [3] | Conversion (%) | Optical Purity (% *ee*) |
|---|---|---|---|
| *N*-Benzoyl-2-MPI | 100 | 21 (48 h) [5] | 63 (*S*) |
| *N*-Pivaloyl-2-MPI | 102 | 31 (48 h) [5] | 88 (*S*) |
| *N*-Acetyl-2-MPI | 2480 | 46 (2 h) [5] | 43 (*S*) |
| *N*-Crotonoyl-2-MPI | 1210 | 56 (4 h) [5] | 55 (*S*) |
| *N*-Benzoylpiperidine | 81200 | n.d. [4] | n.d. |
| *N*-Benzoyl-3-MPI | 7640 | 50 (2 h) [5] | 0 |
| *N*-Pivaloyl-3-MPI | 3120 | 42 (24 h) [5] | 0 |
| *N*-Benzoyl-4-MPI | 9320 | n.d. | n.d. |
| *N*-Benzoylpyrrolidine | 1280 | n.d. | n.d. |
| *N*-Benzoyl-2-methylpyrrolidine | 409 | 49 (24 h) [5] | 11 (*R*) |
| *N*-Benzoyl-2-methylindoline [2] | 40 | 53 (48 h) [5] | 50 (n.d. [4]) |
| *N*-Acetyl-2-methylindoline [2] | 101 | 56 (3 h) [5] | 45 (n.d.) |
| *N*-Pivaloyl-2-methylindoline [2] | 0 | n.d. | n.d. |
| *N*-Acetyl-1,2,3,4-tetrahydroquinaldine [2] | trace | n.d. | n.d. |
| *N*-Benzoyl-1,2,3,4-tetrahydroquinaldine [2] | 0 | n.d. | n.d. |

[1] The reaction was performed at 30 °C in 100 mM of potassium phosphate buffer (pH = 7.0) containing 10 mM of substrate and 0.0158 mg mL$^{-1}$ of purified enzyme. [2] 1 mM of substrate instead of 10 mM was added to the reaction with 5% (*v/v*) acetonitrile. [3] 100% = 0.123 μmol min$^{-1}$ mg$^{-1}$. [4] n.d. = not determined. [5] Reaction time in brackets.

### 2.5. (S)-2-MPI Synthesis Using Recombinant Cells

Since the SHA gene sequence has GC-content, we overexpressed the gene in *Rhodococcus erythropolis* L88, which are high GC-content bacteria that can express high GC-content genes [34]. *Rhodococcus* sp. cells are robust and show resistance to various stress conditions, expecting tolerance to organic solvent, high concentration of substrates, and long-time reaction [35]. The reaction with 100 mM of *N*-pivaloyl-2-MPI using the recombinant cells reached 48.4% conversion to (*S*)-2-MPI with 83.5% *ee* (*E* = 26.3) in 74 h (Figure 3, closed circles). Compared with the reaction using *Arthrobacter* sp. K5 cells that almost stopped after 60 h with the conversion level of <40% (Figure 3, closed triangles), *R. erythropolis* transformant retained the hydrolase activity and the reaction proceeded to 50% conversion after 72 h. Compared to the two-step preparation using the *Aspergillus* sp. protease and acylation of the racemic cyclic amine by the lipase with 3-methoxyphenyl allyl carbonate (Scheme 1B,C), the kinetic resolution by *Arthrobacter* sp. K5 hydrolase achieved one-step preparation of (S)-2-MPI requiring only inexpensive reagents. However, optical purity of 2-MPI was decreased due to gradual hydrolysis of the (*R*)-enantiomer. For the kinetic resolution of chiral 2-MPI production, it is essential to transfer the higher stereoselectivity toward *N*-pivaloyl-2-MPI to SHA using protein engineering.

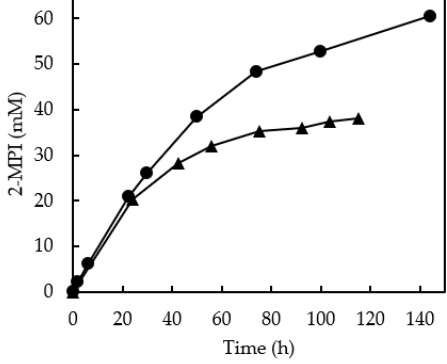

**Figure 3.** Synthesis of (*S*)-2-MPI from *N*-pivaloyl-2-MPI by *Arthrobacter* sp. K5 cells (closed triangles) or recombinant cells (closed circles). The reactions were performed at 30 °C in 100 mM of potassium phosphate buffer (pH = 7.0) containing 100 mM of *N*-pivaloyl-2-MPI and whole cells derived from the culture broth equivalent to a five-fold amount of the reaction volume.

## 3. Materials and Methods

### 3.1. General Information

Commercially available reagents were used without purification and purchased from Tokyo Chemical Industry Co. Ltd. (Tokyo, Japan), FUJIFILM Wako Pure Chemical Corporation (Osaka, Japan) and Sigma-Aldrich (Darmstadt, Germany) unless stated otherwise. The following products from each supplier were used: polypeptone (Nippon Seiyaku, Tokyo, Japan), meat extract (Kyokuto Seiyaku, Tokyo, Japan), and yeast extract (Oriental Yeast, Tokyo, Japan). Thin-layer chromatography (TLC) was performed on TLC silica gel 60F$_{254}$ (Merck KGaA, Darmstadt, Germany). Column chromatography was performed on Wakosil® 60 (FUJIFILM Wako Pure Chemical Corporation, spherical, 64–210 μm). HPLC analyses were performed using LC-10AT pump, SPD-10A detector (Shimadzu, Kyoto, Japan), Atlantis dC18 5 μm 4.6 × 150 mm column (Waters, Massachusetts, USA), CHIRALPAK AD-H 4.6 × 250 mm column (Daicel, Osaka, Japan), and TSK-GEL G-3000SW column (7.5 × 600 mm; Tosoh, Tokyo, Japan). The conversion rate and optical purity were determined by HPLC after derivatization of amines with 2,3,4,6-tetra-*O*-acetyl-β-D-glucopyranosyl isothiocyanate (GITC) at 40 °C for 1 h. The *E* value was calculated using Chen's equation [36]. $^{1}$H and $^{13}$C NMR spectra were recorded on a JEOL ECA600 spectrometer (600 MHz for $^{1}$H and 150 MHz for $^{13}$C) in CDCl$_3$ or CD$_3$OD using tetramethylsilane as an internal standard (δ = 0 ppm). Polymerase chain reaction (PCR), restriction enzyme digestion, and DNA ligation were performed using TaKaRa PCR Thermal Cycler Dice® mini (Takara Bio, Shiga, Japan).

### 3.2. N-Pivaloyl-2-MPI Synthesis

A solution of pivaloyl chloride (13 mL, 107 mmol) in acetonitrile (10 mL) was added to a solution of 2-methylpiperidine (12 mL, 102 mmol) and triethylamine (14.2 mL, 102 mmol) in acetonitrile (60 mL) at 0 °C with stirring. The reaction mixture was warmed to room temperature and stirred overnight. The reaction mixture was filtered to remove triethylamine hydrochloride and the resulting supernatant was concentrated under reduced pressure with a rotary evaporator (EYELA, Tokyo, Japan). The residue was purified by column chromatography (silica gel, *n*-hexane:ethyl acetate = 6:1) to yield *N*-pivaloyl-2-methylpiperidine (17.06 g, 91%) as a colorless oil. $^{1}$H NMR configurations were as follows: CDCl$_3$ (600 MHz), δ (ppm) 1.19, (3H, brs), 1.27 (9H, s), 1.36–1.43 (1H, m), 1.51–1.53 (1H, m), 1.59–1.71 (4H, m), 2.93 (1H, brs), 4.18 (1H, brs), and 4.73 (1H, brs). $^{13}$C NMR configurations are as follows: CDCl$_3$ (150 MHz), δ (ppm) 15.84, 18.88, 26.06, 27.11, 27.89, 28.43, 30.11, 38.83, and 176.12.

### 3.3. N-Acyl Cyclic Amines Synthesis

Acyl chloride (50 mmol) was added to a solution of cyclic amine (50 mmol) and pyridine (50 mmol) in dichloromethane (200 mL) at 0 °C. The reaction was performed overnight at room temperature with stirring. The reaction mixture was concentrated under reduced pressure. Ethyl acetate was added to the residue, and pyridine hydrochloride was removed by filtration. The supernatant was concentrated under reduced pressure and purified by column chromatography (silica gel, *n*-hexane:ethyl acetate = 6:1) to obtain *N*-acyl cyclic amines in moderate to good yield.

### 3.4. Hydrolysis of N-Pivaloyl-2-Methylpiperidine Using Whole Cells of Arthrobacter sp. K5

The reaction was performed at 30 °C with shaking (120 rpm) in 25 mL of 100 of mM potassium phosphate buffer (pH = 7.0) containing 100 mM of *N*-pivaloyl-2-MPI and whole cells derived from 125 mL of culture broth. Samples were collected multiple times and analyzed by HPLC after derivatization of samples with GITC.

### 3.5. Substrate Specificity of Purified Enzyme

Substrate specificity was investigated using the following 1 or 10 mM of *N*-acyl cyclic amines: *N*-benzoyl-2-MPI, *N*-pivaloyl-2-MPI, *N*-acetyl-2-MPI, *N*-acryloyl-2-MPI, *N*-crotonoyl-2-MPI, *N*-benzoylpiperidine, *N*-benzoyl-3-MPI, *N*-pivaloyl-3-MPI, *N*-

benzoyl-4-MPI, *N*-benzoylpyrrolidine, *N*-benzoyl-2-methylpyrrolidine, *N*-benzoyl-2-methylindoline, *N*-acetyl-2-methylindoline, *N*-pivaloyl-2-methylindoline, *N*-acetyl-1,2,3,4-tetrahydroquinaldine, *N*-benzoyl-1,2,3,4-tetrahydroquinaldine, and *N*-acetyl amino acids.

### 3.6. Genome Sequence of Arthrobacter sp. K5

*Arthrobacter* sp. K5 cells were lysed at room temperature overnight with 0.5 mg mL$^{-1}$ of achromopeptidase (crude) in 10 mM of Tris-HCl (pH = 8.0). To this solution, 0.02 mg mL$^{-1}$ of proteinase K, 10 mM of CaCl$_2$, and 0.5% (*w/v*) SDS were added, and the mixture was incubated overnight at 37 °C. To an equal volume of the lysed cells, 2×CTAB solutions were added and incubated at 60 °C for 1 h. The 2×CTAB solution contained 20 g L$^{-1}$ of cetyltrimethylammonium bromide, 50 mM of Tris-HCl, 20 mM of EDTA, 111 g L$^{-1}$ of NaCl, and 10 g L$^{-1}$ of polyvinylpyrrolidone in distilled H$_2$O. To an equal volume of the treatment solution, a mixture of phenol, chloroform, and isoamyl alcohol (25:24:1, *v/v/v*) was added, mixed gently, and centrifuged at 4 °C and 8000 rpm for 30 min. The supernatant was washed with chloroform and one-tenth the volume of 3 M of sodium acetate (pH = 5.2) was added and mixed with isopropanol until genome DNA was thoroughly precipitated. The genome DNA obtained was washed twice with 70% (*v/v*) ethanol and diluted with distilled H$_2$O. Genome analysis was commissioned to Gifu University NGS service.

### 3.7. Hydrolase Overexpression in Rhodococcus Erythropolis L88

The SHA gene was identified based on the partial amino acid sequence of SHA and genome sequence analysis of *Arthrobacter* sp. K5. The gene sequence was deposited in the DDBJ database under the accession number LC633519. The gene amplification was performed via PCR using the primers 5′-CTATCCATGGCAACGCAGACAGTG-3′ and 5′-TAATCTCGAGTCAGACGTTGTCGTCGAGG-3′ with PrimeSTAR$^{®}$ Max DNA Polymerase (Takara Bio). The amplified DNA fragments and pTipQC1 vector (Hokkaido System Science, Hokkaido, Japan) were digested with *Nco* I and *Xho* I and ligated using a DNA Ligation Kit (Takara Bio). The resulting plasmid was transformed into *Rhodococcus erythropolis* L88 cells (Hokkaido System Science) by electroporation with Eporator$^{®}$ (Eppendorf, Hamburg, Germany). The transformed cells were cultivated with 30 µg mL$^{-1}$ of chloramphenicol at 120 rpm and 28 °C in 5 mL of the nutrient medium containing 10 g L$^{-1}$ of tryptone, 5 g L$^{-1}$ of yeast extract, 4 g L$^{-1}$ of Na$_2$HPO$_4$, 1 g L$^{-1}$ of KH$_2$PO$_4$, 1 g L$^{-1}$ of NaCl, 0.2 g L$^{-1}$ of MgSO$_4$·7H$_2$O, and 0.01 g L$^{-1}$ of CaCl$_2$·7H$_2$O in tap water. The preculture was inoculated into 90 mL of the nutrient medium and 0.2 µg ml$^{-1}$ of thiostrepton was added and incubated at 20 °C and 120 rpm for 24 h. The cells were harvested by centrifugation, washed twice with 0.85% (*w/v*) NaCl, and suspended in the same solution.

### 3.8. (S)-2-MPI Synthesis Using Recombinant Cells

The reaction was performed at 30 °C with shaking (120 rpm) in 2 mL of 100 mM of potassium phosphate buffer (pH = 7.0) containing 37 mg (100 mM) of *N*-pivaloyl-2-MPI and whole cells obtained from 10 mL of culture broth. Samples were collected multiple times and analyzed by HPLC after derivatization with GITC.

## 4. Conclusions

SHA exhibited high (*S*)-selectivity toward *N*-pivaloyl-2-MPI to produce (*S*)-2-MPI with 80.2% *ee*. We successfully overexpressed the SHA gene in *R. erythropolis* L88 and improved the hydrolase activity. The biocatalytic process achieved a kinetic resolution of 100 mM of *N*-pivaloyl-2-MPI in one step using inexpensive synthetic substrates, forming (*S*)-2-MPI with 48.4% conversion and 83.5% *ee*. As a potential enzyme for practical applications, SHA may allow highly enantioenriched (*S*)-MPI production by improving its stereoselectivity in the future.

**Supplementary Materials:** The following are available online at https://www.mdpi.com/article/10.3390/catal11070809/s1, Figure S1: NMR spectrum of *N*-pivaloyl-2-MPI, Table S1: Summary of purification steps of the enzyme, Figure S2: SDS-PAGE of the enzyme, Figures S3–S6: Effects of temperature and pH on enzyme activity, Table S2: HPLC analysis conditions, Figures S7–S9: HPLC chromatograms.

**Author Contributions:** Conceptualization, T.Y. and K.M.; synthesis of substrates, K.K., Y.M. and Y.F.; screening of microorganism and optimization of culture conditions, K.K.; purification of enzyme and substrate specificity, Y.M.; construction of expression system, Y.F.; writing—original draft preparation, Y.F.; writing—review and editing, Y.F., T.Y. and K.M.; supervision, T.Y. and K.M. All authors have read and agreed to the published version of the manuscript.

**Funding:** This research received no external funding.

**Data Availability Statement:** All data are available in the main text, the supplementary materials or DDBJ database.

**Acknowledgments:** We thank Toru Nagasawa for directing this research. The authors are grateful to the Life Science Research Center, Gifu University, for NMR analysis and whole-genome sequencing. The authors would like to thank Enago (www.enago.jp) for the English language review.

**Conflicts of Interest:** The authors declare no conflict of interest. The funders had no role in the design of the study; in the collection, analyses, or interpretation of data; in the writing of the manuscript, or in the decision to publish the results.

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
