# Peer review of "Novel (S)-Selective Hydrolase from Arthrobacter sp. K5 for Kinetic Resolution of Cyclic Amines"

_catalysts, doi:10.3390/catal11070809_

Round 1

Reviewer 1 Report

The authors describe the identification of a strain of Arthrobacter that contains a hydrolase capable of hydrolysing an N-pivaloyl cyclic amine with acceptable enantioselctivity.  The protein was purified the gene cloned, and then expressed in Rhodococcus.  The recombinant strain was also competent for the resolution reaction. The observation is interesting, but the presentation of the work is not sufficiently clear – a lot of data and experimental details are missing, there are errors in presentation in figures and tables and the reasons for  using Rhodococcus are not made clear. 

Section 2.2 ‘the best compound’; the highest activity’ ‘decreases enzyme activity’ really need to provide some data here.

Page 3  ee or e.e. Not EE

What is meant by the last sentence of Section 2.2? The ‘wild-type cells’ appear to catalyze the reaction quite well; if the E value could be calculated, this would give the measure of ‘efficiency’ of ‘kinetic resolution’

Section 2.3 – Why do you present approximations of molecular weight  - 50 kDa monomer and a ‘238 kDa’ tetramer, when the full sequence of the gene will tell you what the true MW is?

Section 2.3  - what is meant by 67% homology? ID? Similarity?

Table 1  - how were ees measured?  How were absolute configurations assigned?

Were the ees of residual subsrates not measured? This wold permit calculation of the E value. Why is the NMR of only one substrate presented, and not the other substrates or the products?  The data are not comprehensive.

The table header for the % ee column is wrong as the bracket value in the table is not the e.e as suggested.

Section 2.5 – not really clear why the gene was expressed in Rhodococcus? What was the advantage?  Why not use E. coli?

Section 2.5 – How was the gene expressed in R. erythropolis?  This is not a standard technique – please give details.  No data or evidence of the gene expression in the organism are provided – the gel is in the SI, but the details of cell disruption, purification and chromatography are not.  Also the gel could not possibly show the cell extract; the protein looks almost pure.

In the experimental, the growth times for a Rhodococcus strain are surprisingly short (24 h)

Remove ‘the solution was used as whole cells’ – this does not make sense

Results and conclusion, abstract – please include % isolated yield of the products

Several errors in the SI – graphs that go up to 120%; ‘erythropolis’ typos etc. please check

Author Response

Thank you very much for your useful comments and accurate remarks. We have made corrections and additions to the paper based on your comments.

The following corrections and additions have been made to the points pointed out. We hope you can confirm this.

Comment 1: Section2.2 ’the best compound’; the highest activity’ ‘decreases enzyme activity’ really need to provide some data here.

Response: Table S1-S3 about the detail of optimization has been added to supporting materials.

Comment 2: Page 3 ee or e.e. Not EE

Response: Thank you for pointing this out. We corrected it.

Comment 3: What is meant by the last sentence of Section 2.2? The ‘wild-type cells’ appear to catalyzed the reaction quite well; if the E value could be calculated, this would give the measure of ‘efficiency’ of ‘kinetic resolution’

Response: Because we aim 50% conversion of racemic substrate, we think the productivity of (S)-2-MPI should be improved. We added the E value of (S)-MPI prepared from N-pivaloyl-2-MPI (wild-type cells, E = 14.8; recombinant cells, E = 26.3). In this paper, we mainly focus the property of the hydrolase, and will study the stability of the hydrolase. After enzyme improvement (variant construction), we will synthesize various chiral cyclic amines.

Comment 4: Why do you present approximations of molecular weight – 50 kDa monomer and ‘238 kDa’ tetramer, when the full sequence of the gene will tell you what the true MW is?

Response: Thank you for pointing this out. We added “The enzyme gene was composed of 1,384 bp, and coded for a protein of 481 amino acids with a molecular mass of 49,725 Da. This value is in agreement with molecular mass determined on SDS-PAGE.” to section 2.3.

Comment 5: what is meant by 67% homology? ID? Similarity?

Response: Thank you for pointing this out. It’s sequence identity. We replace ‘homology’ with ‘sequence identity’.

Comment 6: How were ees measured? How were absolute configurations assigned?

Response: We assigned absolute configuration using commercially available chiral cyclic amines. We added ‘Absolute configurations of chiral cyclic amines were assigned using commercially available chiral reagents.’ under Table S5.

Comment 7: Were the ees of residual substrates not measured? This would permit calculation of the E value. Why is the NMR of only one substrate presented, and not the other substrates or the products? The data are not comprehensive.

Response: We didn’t measure the ees of residual substrate. NMR data were added to supporting materials.

Comment 8: The table header for the % ee column is wrong as the bracket value in the table is not the ee as suggested.

Response: Thank you for pointing this out. We revised the table.

Comment 9: Section 2.5 – not really clear why the gene was expressed in Rhodococcus? What was the advantage? Why not use E. coli?

Response: We recently try recombinant Rhocococcus to expect its tolerance to organic solvent and high concentration of substrate. We added ‘Rhodococcus sp. cells are robust and show resistance to various stress conditions, expecting tolerance to organic solvent, high concentration of substrates, and long-time reaction [35].’ to section 2.5 with a reference.

Comment 10: Section 2.5 – How was the gene expressed in R. erythropolis? This is not a standard technique – please give details. No data or evidence of the gene expression in the organism are provided – the gel is in the SI, but the details of cell disruption, purification and chromatography are not. Also the gel could not possibly show the cell extract; the protein looks almost pure.

Response: The method of gene expression was described in section 3.7. The cell-free extract used for SDS-PAGE in supporting materials was prepared just by disruption of the cells, not using chromatography. We were also surprised at this result. We replaced the picture of SDS-PAGE with the high-concentration version in which other proteins can be seen.

Comment 11: In the experimental, the growth times for a Rhodococcus strain are surprisingly short (24 h)

Response: The time ‘24 h’ is incubation time for induction of the hydrolase gene. Precultivation before induction is longer. It takes 2-3 days.

Comment 12: Remove ‘the solution was used as whole cells’ – this does not make sense.

Response: Thank you for pointing this out. This sentence was removed.

Comment 13: Results and conclusion, abstract – please include % isolated yield of the products.

Response: We didn’t calculate isolated yield of the product. As we described in comment 3, this study mainly focus the property of the hydrolase. We will discuss preparation of chiral 2-MPI next paper after enzyme engineering.

Comment 14 : Several errors in the SI – graph that go up to 120%; ‘erythropolis’ typos etc. please check.

Response: Thank you for pointing this out. We checked mistakes as far as possible.

Reviewer 2 Report

This manuscript focuses on the stereoselective hydrolases for 2-MPI production. This is an interesting research and the results seem to be promising. My only concern is that I didn't see the significance of this work. How do you compare with other similar work? Do you have better performance or comparable performance? What are the advantages of your methods? I'd like to see more explanations in the introduction and discussions sessions to show that your work has some contributions to this research field, instead of simply presenting your own results.

Author Response

Thank you very much for your useful comments and accurate remarks. We have made corrections and additions to the paper based on your comments.

The following corrections and additions have been made to the points pointed out. We hope you can confirm this.

Comment: This manuscript focuses on the stereoselective hydrolases for 2-MPI production. This is an interesting research and the result seem to be promising. My only concern is that I didn’t see the significance of this work. How do you compare with other similar work? Do you have better performance or comparable performance? What are the advantages of your methods? I’d like to see more explanations in the introduction and discussions sessions to show that your work has some contributions to this research field, instead of simply presenting your own results.

Response: Thank you for pointing this out. We added the ‘scheme 1’ to explain the reported process for kinetic resolution of cyclic amines. We also added some sentences in the manuscript as follow:

Section 1 (introduction), paragraph 2, line 15.

“Interestingly, there is no approach to directly obtain chiral 2-substituted piperidines from racemic N-acyl derivatives by kinetic resolution using a hydrolase.” was added.

Section 1 (introduction), paragraph 3, line 2.

“for one-step preparation of chiral 2-MPI (Scheme 2)” was added.

Section 2.5, line 9.

“Compared to two-step preparation using the Aspergillus sp. protease and acylation of the racemic cyclic amine by the lipase with 3-methoxyphenyl allyl carbonate (Scheme 1, B and C), the kinetic resolution by Arthrobacter sp. K5 hydrolase achieved one-step preparation of (S)-2-MPI requiring only inexpensive reagents.” was added.

Section 4 (conclutions), line4.

in one step using inexpensive synthetic substrates.